# A Comparative Phytochemical Investigation of the Greek Members of the Genus *Helichrysum* Mill., with Emphasis on the Local Endemic *Helichrysum amorginum* Boiss and Orph

**DOI:** 10.3390/plants14020229

**Published:** 2025-01-15

**Authors:** Iordanis Samanidis, Nikos Krigas, Vassilis Athanasiadis, Ioannis Makrygiannis, Martha Mantiniotou, Stavros I. Lalas

**Affiliations:** 1Department of Food Science and Nutrition, University of Thessaly, Terma N. Temponera Str., 43100 Karditsa, Greece; isaman@uth.gr (I.S.); vaathanasiadis@uth.gr (V.A.); ioanmakr1@uth.gr (I.M.); mmantiniotou@uth.gr (M.M.); 2Institute of Plant Breeding and Genetic Resources, Hellenic Agricultural Organization-Demeter (ELGO-Dimitra), 57001 Thermi-Thessaloniki, Greece; nkrigas@elgo.gr; 3Department of Viticulture, Floriculture & Plant Protection, Institute of Olive Tree, Subtropical Crops and Viticulture, Hellenic Agricultural Organization-Demeter (ELGO-Dimitra), 71307 Heraklion, Greece

**Keywords:** PLE, LC-HRMS, RP-HPLC-DAD, total polyphenols, neochlorogenic acid, chlorogenic acid, astragalin, non-volatile secondary metabolites

## Abstract

The members of the genus *Helichrysum* Mill. are notable for producing a diverse range of structurally intricate secondary metabolites, being the focus of current phytochemical research. Their importance is recognized as several species hold significant ethnopharmacological value, being traditionally used to address ailments in human systems, such as respiratory, gastrointestinal, and urinary conditions, among others. This study used liquid chromatography coupled with high-resolution mass spectrometry results to present the phytochemical composition of non-volatile secondary metabolites in 11 Greek *Helichrysum* taxa (species and subspecies). For the first time, their total polyphenol content is comparatively assessed and an overview of the non-volatile compounds for five Endangered or Critically Endangered Greek Endemic *Helichrysum* taxa are presented herein. Almost all of the studied *Helichrysum* taxa differed significantly in the mean values of their polyphenolic content, except for *H. doerfleri* and *H. heldreichii*. A particular emphasis was placed on extracting polyphenols from a cultivated genotype of *H. amorginum* using aqueous pressurized liquid extraction as an alternative to the traditional organic solvent extraction method. Analysis by high-performance liquid chromatography revealed that this method increased the concentration of neochlorogenic acid and astragalin, compared to other extraction conditions. These findings highlight the potential of alternative extraction techniques for obtaining natural products from sustainably managed phytogenetic sources.

## 1. Introduction

The genus *Helichrysum* Mill. (Asteraceae) includes at least 559 accepted species worldwide (https://powo.science.kew.org/taxon/urn:lsid:ipni.org:names:331648-2, accessed on 11 December 2024). In Greece, this genus includes 11 taxa (species and subspecies) which are classified into three sections, i.e., *Helichrysum*, *Stoechadina* (DC.) Gren. and Godr., and *Virginea* (DC.) Fiori [1,2]. From a phytogeographic viewpoint, the globally widespread *H. luteoalbum* (L.) Rchb. is distributed in 9 of the 13 phytogeographic regions of Greece (https://portal.cybertaxonomy.org/flora-greece/intro, accessed on 11 December 2024), similarly to the Mediterranean *H. stoechas* (L.) Moench subsp. *barrelieri* (Ten.) Nyman (HSB), while others, such as *H. italicum* subsp. *microphyllum* (Willd.) Nyman (HIM) [3,4], *H. plicatum* (HP), and *H. orientale* (HO), are of eastern Mediterranean range. In total, five species are local Greek endemics with range-restricted distribution [3,4], namely the Endangered [5] *H. amorginum* Boiss. and Orph. (HA), found in some Cyclades Islands (Amorgos, Keros, Anydros, Folegandros, Anafi), and *H. taenari* Rothm. (HT) in south Peloponnese, as well as the Critically Endangered [5] *H. sibthorpii* Rouy (HS), found at the summit of Mount Athos, *H. doerfleri* Rech. f. (HD) in the Mount Thrypti summit area of eastern Crete, and *H. heldreichii* Boiss. (HH) in some gorges of south-western Crete. Among the range-restricted [3,4] and threatened [5] Greek endemic taxa (additional information about the phytogeographical distribution of the studied *Helichrysum* taxa in Greece, local endemism, and extinction risk status is presented in Appendix A), *H. amorginum* is found in the Cyclades (*Kik)* floristic area, in Amorgos island (north and central part of the island, at an altitude of 200–700 m above sea level, and in the area of the monastery of Hozoviotissa with south and south-east exposure), as well as on the islets of Anydros (Amorgopoula) and Keros, with additional specimens of smaller populations being found on the Anafi and Folegandros islands [6]. This species (locally called ‘stathouri’ or ‘amorgino helichryso’ in Greek, especially on Amorgos Island) flowers in March–April (flowering initiation), is usually in full bloom in May(–June), and by July (end of flowering), fruit formation is complete. It is a rare hemicryptophyte, growing in the crevices of limestone rocks as a chasmophyte; being always found in small populations with scattered individuals, it is assessed as an Endangered species [5]. This species morphologically employs perennial plants with grey-to-white indumentum and erect stems 10–30 cm long, with anti-lanceolate leaves (up to 40 mm) in clusters and impressive compound inflorescences (terminal corymb of 6–15 heads with yellow tubular florets and oblong, blunt, loosely overlapping papery bracts that are rose-pink-to-pink in bud and white in flowering) [4]. It is a diploid species with a chromosome number of 2n = 28 [7].

In modern phylogenetic terms, the current literature has mainly focused on *Helichrysum* taxa that are not found in Greece, and only one study has included the Greek endemic *H. sibthorpii* [8]. From a systematic viewpoint, taxonomically closely related taxa are *H. amorginum* Boiss. and Orph., *H. doerfleri* Rech. f., *H. sibthorpii* Rouy, and *H. taenari* Rothm. [4]; *H. italicum* (Roth) G. Don subsp. *italicum* (HII) is related to *H. stoechas* subsp. *barrelieri* [4]; *H. luteoalbum* (HL) is related to the African *H. reflexum* N.E.Br. [4]; and *H. heldreichii* is related to *H. italicum* and *H. stoechas* [4].

From a pharmacognostic viewpoint, the reported studies of members of the genus *Helichrysum* include investigations of various secondary metabolites, such as phenolic acids and derivatives, flavonoids, terpenes, pyrones, phloroglucinols, and benzofurans [9]. Several pharmacological properties have also been reported investigating the capability of *Helichrysum* species’ extracts to suppress inflammation, the growth of pathogenic microorganisms, and the inhibition of enzymes [9]. In the context of ethnobotany, some species of the genus are traditionally used as therapeutic agents. For example, *H. stoechas* subsp. *barrelieri* has been used as a component of a herbal tea mixture called ‘Zahraa’ that is consumed as a digestive [10]; *H. petiolare* Hilliard and B.L. Burtt and *H. odoratissimum* (L.) Sweet have been used as constituents of the traditional drug ‘Imphepho’, used in tropical Africa to treat respiratory and urinary infections [11]; and *H. arenarium* (L.) Moench has been used to treat gastrointestinal disorders [12]. Modern uses of *Helichrysum* species are related mainly to *H. italicum*, and extend to the application of its essential oil in cosmetics [13], the use of its extracts in food supplements [14], and the use of its isolated bioactive secondary metabolites, such as arzanol [15].

Prominent phytochemical studies on the Greek *Helichrysum* taxa were initially conducted regarding their essential oils, and date back to the early works on *H. amorginum* and *H. italicum* subsp. *Italicum*, in which the bacteriostatic activity of the oils was also assessed [16]; *H. stoechas* subsp. *barrelieri* and *H. taenari*, with complementary tests on their antibacterial activity [17]; *H. doerfleri*, *H. heldreichii*, *H. italicum* subsp. *microphyllum*, and *H. orientale*, with bacteriostatic activity tests on their oils and pentane-dichloromethane extracts [18]; *H. stoechas* subsp. *barrelieri*, with antimicrobial tests on the efficacy of its oil against bacteria and pathogenic fungi [19]; and *H. amorginum*, with antimicrobial tests on the efficacy of its oil against bacteria and pathogenic fungi [20]. *Helichrysum sibthorpii*, however, has not yet been investigated to date, and there is a complete absence of phytochemical studies regarding the non-volatile metabolites of the Greek endemic *Helichrysum* taxa. Relative phytochemical studies exist only for species with a wider distribution, namely, *H. luteoalbum* [21], *H. stoechas* subsp. *barrelieri* [22], *H. italicum* subsp. *microphyllum* [23], *H. italicum* subsp. *italicum* [24], *H. plicatum* [25], and *H. orientale* [26]. Regarding *H. amorginum* specifically, only one previous publication has studied the polyphenol content of this species’ methanol extracts after domestication of wild-growing biotypes and their sustainable cultivation [27], and quite recently, a doctoral thesis was completed on the phytochemical investigation of the 11 Greek *Helichrysum* taxa, focusing especially on *H. amorginum* [28]. Nevertheless, the bibliography still lacks information on the phytochemistry and pharmacological properties of the non-volatile constituents of *H. amorginum*.

Context-wise, the present work aims to highlight the non-volatile phytochemical profile of the 11 wild-growing *Helichrysum* taxa of Greece, comparatively assessing their total polyphenol content and paving the way for the utilization mainly of the threatened local endemic ones, in the frame of sustainable exploitation strategies. Moreover, this study explores the potential of an alternative extraction technique to recover valuable phytochemical compounds by eliminating the use of organic solvents from an ex situ cultivated genotype of *H. amorginum* that originated from wild-growing biotypes, thus providing novel insights into this local Greek endemic species, which is threatened with extinction, when being sustainably maintained in ex situ cultivation.

## 2. Results and Discussion

### 2.1. Untargeted Analysis of Secondary Metabolites by LC-HRMS/MS

The phytochemical profile of the MeOH extracts (Table 1) of the 11 Greek *Helichrysum* taxa (ten taxa of wild collection, and the cultivated genotype of *H. amorginum*, originating directly from wild plant material) has been analysed in a previous study utilizing a UPLC-HR(ESI+/−)MS/MS technique [28].

LC-HRMS is a powerful tool in the natural products screening of various plant species, and similar studies have been conducted for the MeOH extracts of the local endemics of Madeira Island, namely *H. devium* J.Y.Johnson [29], *H. melaleucum* Rchb. ex Holl [30], *H. monizii* Lowe [31], *H. obconicum* DC. [32], and the EtOH extract of *H. stoechas* subsp. *barrelieri* [22].

According to Table 1, all the MeOH extracts showed a metabolite profile that is typical of the Greek members of the genus *Helichrysum*, a trend that aligns with ones previously described in a recent comprehensive review [9]. More specifically, the extracts were found to contain a variety of known and unidentified metabolite structures with differing polarities. The reported metabolites could be categorized into four groups, i.e., acids and derivatives, flavonoids, pyrones and derivatives, and others. Examples of polar-to-medium-polar compounds include sugars, quinic acid, malic acid, benzoic acid derivatives, hydroxycinnamic acid derivatives (caffeic acid derivatives), and coumarins (aesculin, aesculetin). In addition to this, a series of flavonoid classes, like flavonols (kaempferol, quercetin), flavanones (naringenin), and flavones (apigenin, luteolin) in their aglycone, glycosidic, or more complex form, were also found to be present in the materials studied herein. Examples of medium-polarity-to-apolar compounds include methylated flavonoids (5,7-dihydroxy-3-methoxy flavone, isorhamnetin), terpenes, and hydroxy fatty acids. It is worth mentioning a wide series of pyrone derivatives and analogues, which range from monomeric pyrones (micropyrone) to heterodimeric phloroglycinyl pyrone (arzanol) and acetophenone derivatives (italipyrone). Nonetheless, the profiles of the MeOH extracts for the Endangered local endemics *H. amorginum* and *H. taenari*, and those of the Critically Endangered *H. doerfleri, H. heldreichii*, and *H. sibthorpii*, are presented for the first time in this investigation. Although pivotally investigated herein, and based on limited quantities of wild-growing plant material, such data can serve as reference baseline information for the future ex situ cultivation of these range-restricted and threatened taxa [3,4,5]. Undoubtedly, for any in-depth future investigation, isolation, and characterization of specific compounds or possible industrial applications, further research is needed that employs comparative studies of ex situ cultivated genotypes of wild origin.

As far as *H. amorginum* is concerned, the sustainably ex situ cultivated genotype that originated directly from the wild was found to contain several caffeic acid derivatives, like mono- and di- caffeoyl quinic acid isomers; several kaempferol glycosides, notably astragalin and tiliroside; hydroxy fatty acids; pyrone derivatives, especially arzanol, 3-methylarzanol, heliarzanol, cycloarzanol, arenol, and italipyrone; unknown pyrone derivatives; unknown prenylated polyphenols; and unknown prenyl phloroglucinol derivatives.

In summary, this preliminary dereplication study on the MeOH extracts from the 11 Greek *Helichrysum* taxa provides new insights into their polyphenolic content, which is further examined in Section 2.2.

### 2.2. Polyphenol Extraction from Greek Helichrysum Taxa

Table 2 depicts the results of the polyphenol content of the 11 Greek *Helichrysum* taxa, expressed as the yield of total polyphenols (mg GAE per g of dry weight of plant material) and as mg GAE/L of liquid extract. The polyphenolic content was determined after a hydromethanolic extraction performed by stirring (STE) under the following conditions: MeOH:H_2_O, 7:3, 21–22 °C and the solvent’s static dielectric constant (*ε*) of 42.54 (20 °C) [33]. All the *Helichrysum* taxa showed statistically significant differences between the mean values (mg GAE/g dw) of their polyphenolic content, except HD and HH. Higher polyphenol content values were observed in HIM and HP, followed by HSB and HT, whereas medium values were observed in HII, HS, and HA. Lower values were observed in HH, HD, HO, and HL. A similar study [34] focused on *H. arenarium* inflorescences using hot water extraction, and reported a polyphenol content ranging from 61.4 to 92.3 mg/g of plant material (expressed in pyrogallol equivalents); such values are comparable to the levels found herein in the Greek materials of HT, HSB, and HP. Furthermore, in another study [35] on *H. arenarium* using ultrasound-assisted extraction with MeOH:H_2_O (1:1), polyphenol contents of 1062.82 ± 12.36 mg GAE/L (leaf extract) and 652.56 ± 5.87 mg GAE/L (flower extract) were reported. These results highlight the effect of different extraction solvents and parts of plant material used to extract polyphenols from different *Helichrysum* species.

The results of the polyphenol content of the HA genotype under different extraction conditions are depicted in Table 3. The results are expressed as the yield of total polyphenols (mg GAE per g of the dry weight of plant material) and as mg GAE/L liquid extract. All the mean values of total polyphenol content were statistically different, except for MeOH (STE) and H_2_O (STE). It was observed that a higher yield of polyphenols was achieved with MeOH:H_2_O mixtures (9:1 and 7:3) and higher solvent static dielectric constants (36.1, and 42.54, respectively) [33]. The highest yield was achieved with MeOH:H_2_O (7:3) with *ε* = 42.54. For PLE water extraction with *ε* = 39.5 [36], the yield was also close to that of the MeOH:H_2_O (7:3) mixture. These results indicate that the static dielectric constant may be used as an auxiliary reference parameter to adjust the conditions of PLE water extraction, achieving polyphenol yields comparable to those of hydroethanolic mixtures.

Known studies regarding *H. amorginum* total polyphenols are limited to only one recent one [27], in which the polyphenol contents in MeOH dry extracts of the cultivated *H. amorginum* were studied during different harvest stages. A polyphenol content in MeOH dry extracts of 142.6–156 mg GAE/g was reported in the early flowering stage, followed by 139–145.7 mg GAE/g in full bloom, and 130.3–131 mg GAE/g in late bloom. However, as these results refer to MeOH dry extracts, they could not be directly compared with those of the current study.

### 2.3. UV-Vis and HPLC Quantification of Major Metabolites from Helichrysum amorginum

As a next step, the UV-vis scanning absorption spectra at 190–700 nm of the HA extracts H_2_O (PLE), H_2_O (STE), and MeOH:H_2_O (7:3, STE) were studied to determine their λ_max_ absorption wavelengths (Figure 1, Table 4). The crude extracts used in this study were not purified; therefore, the presence of unknown impurities may have influenced the UV-vis absorption spectra. These impurities could include soluble proteins, amino acids, and other organic compounds that absorb at similar wavelengths. The presence of impurities can lead to overlapping absorption peaks, making it challenging to accurately attribute the observed *λ*_max_ values to specific compounds.

A similar absorption pattern was observed in the three extracts. More specifically, the aqueous extracts H_2_O (PLE) and H_2_O (STE) showed similar λ_max_ at 266, 286, and 321 nm, and λ_max_ at 266, 289, and 321 nm, respectively (Table 4). On the other hand, the MeOH:H_2_O extract showed a similar λ_max_ at 265, 296, and 322, with a difference in wavelength of 296 vs. 286/289 for the aqueous extracts, which may be attributed to its more powerful solvating effect and selectivity for polar-to-medium-polar compounds. According to the literature [37], the absorption spectrum of flavonoids typically consists of two main bands, band I (300–380 nm) and band II (240–295 nm), with the former causing a yellow colour. For some flavonoids, absorption creates a tail at 400–450 nm. Similarly, the three extracts’ absorbances λ_max_ at 266 nm, 266 nm, and 265 nm are attributed to band II of the A-ring benzoyl system of flavonoids, whereas the λ_max_ at 321 nm, 321 nm, and 322 nm are attributed to band I of the B-ring cinnamoyl system of flavonoids. Additionally, according to sources from the literature [38], compounds with absorption ranges of 246.1–250.0 nm, 324.7–329.3 nm, and with a shoulder at 290–300 nm are attributed to chlorogenic acids. Also, in extracts of *H. italicum* [39], the λ_max_ at 329 nm has been used to quantify caffeoylquinic acids. In the three extracts examined herein, chlorogenic acids could be associated with the shoulder at *λ*_max_ values of 286 nm, 289 nm, and 296 nm. Nonetheless, it should not be neglected that the absorbance ranges and λ_max_ values have a complementary and indicative role in the prediction of compounds’ categories. Other characteristic compound categories may contribute to the three extracts’ absorbances, as in the case of phloroglucinols and tremetones that absorb at 280 nm [39].

The three wavelengths 266 nm, 286 nm, and 321 nm were selected for the initial monitoring of the HPLC-DAD absorbance chromatograms of the three studied extracts (Appendix A).

From the HPLC-DAD chromatograms, one major metabolite, kaempferol-3-*O*-glucoside (astragalin; AST), at Rt = 38.8–38.9 min with λ_max_ = 265 nm and 345 nm, chlorogenic acid (CA; 3-*O*-caffeoylquinic acid) at Rt = 21.8–22.0 min with λ_max_ = 324 nm, and neochlorogenic acid (NCA; 5-*O*-caffeoylquinic acid) at Rt = 16.50–16.54 min with λ_max_ = 324 nm, were identified and quantified at λ_max_ = 265 nm (AST) and λ_max_ = 324 nm (CA, NCA), respectively. For the identification of reference standards, astragalin with Rt = 38.72 min, chlorogenic acid with Rt = 21.95 min, and neochlorogenic acid with Rt = 16.50 min were used. As expected, the peak intensity of astragalin was much higher at 266 nm (Appendix A) and the peak intensities of chlorogenic acid and neochlorogenic acid were higher at 286 and 321 nm (Appendix A). The quantitative results, expressed as extract concentration (mg/L) and as mg/g of dry weight (dw) of plant material, are presented in Table 5.

Regarding the NCA, the highest concentration of 130.91 mg/L was achieved in the H_2_O (PLE) extract, corresponding to 2.61 mg/g of dry plant material content. A higher concentration of 24.41 mg/L was observed in the MeOH:H_2_O (7:3, STE), while a lower concentration of 19.83 mg/L was detected in the H_2_O (STE). From these findings, H_2_O (PLE) positively affected neochlorogenic acid extraction. The NCA content was also measured in *H. italicum* herb (aerial parts) and inflorescences after an ultrasound-assisted methanol extraction [40]. Low NCA levels of 0.8583 mg/g dw plant material in the herb and 0.5941 mg/g dw plant material in the inflorescences were reported, emphasizing the significance of metabolite accumulation across all aerial parts of the species, not just in the inflorescences.

Regarding CA, the highest concentration of 188.67 mg/L was achieved in the MeOH:H_2_O (7:3, STE) extract, corresponding to 3.76 mg/g of dry plant material. A higher concentration of 134.66 mg/L was observed in H_2_O (STE), while a lower concentration of 115.64 mg/L was detected in H_2_O (PLE). From these findings, H_2_O (PLE) did not drastically affect chlorogenic acid extraction. The CA content has also been measured in *H. italicum* herb (aerial parts) and inflorescences after an ultrasound-assisted methanol extraction [40]. CA levels of 3.3881 mg/g dw plant material in the herb and 1.4294 mg/g dw plant material in the inflorescences were reported, emphasizing again the significance of metabolite accumulation across all aerial parts of the species. It is worth mentioning that the extraction conditions of *H. italicum* aerial parts afforded a comparable CA content (3.3881 mg/g dw) to that of *H. amorginum* aerial parts (3.76 mg/g dw), with MeOH:H_2_O (7:3) as the extraction solvent.

Regarding the AST, the highest concentration of 694.36 mg/L was observed in the MeOH:H_2_O (7:3, STE) extract, which corresponds to a plant material content of 13.85 mg/g. Next, a higher concentration of 206.57 mg/L was observed in H_2_O (PLE), and a lower concentration in H_2_O (STE) of 143.60 mg/L. Based on these findings, H_2_O (PLE) achieved a 43.85% increase in astragalin concentration compared to maceration in water. Astragalin is considered to be a valuable bioactive flavonol [41] that is slightly water-soluble, with mentioned solubility values of 28.2 mg/L in water [42] and 140 mg/L in a 1:6 solution of DMSO:PBS [43]. In this study, the quantification of 143.60 mg/L of AST in the water extract (STE) (Table 5) aligns most closely with the results for a 1:6 DMSO:PBS solution.

Specific studies applying PLE or ASE (Accelerated Solvent Extraction) methods with a focus solely on *Helichrysum* spp. are limited. A 2013 study [44] using ASE for the extraction of *H. italicum* achieved a high yield of 13,375.0 mg of astragalin per kg of dry extract, with MeOH:H_2_O (7.5:2.5) as the solvent. However, in another study on *H. stoechas* subsp. *barrelieri* [22] which compared five extraction techniques with EtOH as the extraction solvent, the ASE (120 °C/1500 psi) yielded 7.016 mg of astragalin per kg of dry extract. The highest yield was achieved by maceration (30.497 mg/kg), followed by ultrasound-assisted extraction (16.025 mg/kg), Soxhlet extraction (4.950 mg/kg), and microwave-assisted extraction (0.830 mg/kg). These results indicate that the extraction yield of astragalin is greatly affected by the extraction technique, extraction parameters (like temperature, solvent, time, etc.), and its initial content in the plant material.

However, the findings of this study on HA are based on a single genotype, and it would be intriguing to conduct comparative investigations involving additional HA genotypes. Furthermore, exploring other extraction preparation conditions, such as plant material drying methods and extraction techniques, is essential to understanding their impact on the species’ phytochemical profile. Additionally, quantifying other secondary metabolites under optimized extraction conditions would provide valuable insights into the potential utilization of the species’ phytochemicals.

## 3. Materials and Methods

### 3.1. Plant Material Collection and Sample Preparation

All the plant material studied in this investigation originated from wild-growing populations of Greek *Helichrysum* taxa; the botanical collections were performed using a special permit (Permit 82336/879, updated on 18 May 2019, and 26895/1527, updated on 21 April 2021) issued by the Greek Ministry of Environment and Energy. The initial plant materials collected from the wild for seven Greek native *Helichrysum* taxa were asexually propagated and ex situ maintained for conservation purposes at the Institute of Plant Breeding and Genetic Resources of the Hellenic Agricultural Organization Demeter (ELGO-Dimitra) in Thermi, Thessaloniki, Greece, with IPEN (International Plant Exchange Network) accession numbers encoded as follows: GR-1-BBGK-20,136 for *H. doerfleri* [HD: fully bloomed wild plant material originally collected by Dr. Iordanis Samanidis (I.S.), on 24 June 2019, from rocky (limestone) outcrops on Mount Thripti, Crete, Greece, at 1312 m above sea level]; GR-1-BBGK-20,134 for *H. heldreichii* [HH: wild plant material originally collected by Ioannis Kofinas-Kallergis (I.K.-K.) and I.S., on 23 June 2019, in Aradaina Gorge, Crete, Greece, at 520 m above sea level]; GR-1-BBGK-20,132 for *H. italicum* subsp. *microphyllum* (HIM: fully bloomed wild plant material originally collected by I.K.-K. and I.S., on 23 June 2019, in a rocky area of Sfakia, Crete, Greece, at 573 m above sea level); GR-1-BBGK-20,133 for *H. orientale* (HO: bloomed wild-growing plant material harvested in 2018 from the islet of Chytra and purchased from a local market in Kythera Island, in May 2019, by I.S.); GR-1-BBGK-20,137 for *H. sibthorpii* [HS: fully bloomed wild plant material originally collected by I.S. and I.K.-K., on 5 August 2019, at the rocky (limestone) summit area (1950–2033 m) of Mount Athos, Monastic community of Mount Athos, Greece]; GR-1-BBGK-20,135 for *H. taenari* [HT: fully bloomed wild plant material originally collected by I.S. and I.K.-K., on 17.05.2019, from a rocky (limestone) outcrop at Cape Taenaro, Peloponnese, Greece]; and GR-1-BBGK-19,726 for *H. plicatum* [HP: bloomed wild plant material originally collected by Dr. Nikos Krigas (N.K.), on 1 June 2020, in rocky habitats at 1426 m above sea level on Mount Varnountas, Florina, Greece]. The genotype studied herein, GR-1-BBGK-15,5871 of *H. amorginum* (HA), originated from wild-growing plant material in full bloom on vertical limestone rocks close to Panagia Hozoviotissa Monastery, Amorgos Island, Greece, originally collected by N.K. and Dr. Georgios Tsoktouridis in early March of 2015. Individually taxonomically identified specimens from these materials were deposited at the herbarium of the Balkan Botanic Garden of Kroussia (BBGK), with respective IPEN accession numbers.

The plant material of *H. stoechas* subsp. *barrelieri* (HSB) was originally collected by I.S. from a rocky (limestone) outcrop at 106 m above sea level on Agios Stefanos Hill (Monte Smith), Rhodes, Greece. The plant material of *H. italicum* subsp. *italicum* (HII) was originally collected by Amorgos Collaboration SCE, in the summer of 2019, in its natural phryganic habitat on Amorgos Island, Greece (the specimen was identified by I.S.). The plant material of *H. luteoalbum* (HL) was identified and originally collected by Dr. Eleftherios Kalpoutzakis, on 25.6.2020, in its natural habitat in an area of Mount Parnonas, Peloponnese, Greece. For these materials, herbarium specimens were kept by I.S.

After collection, all the sampled wild plant materials were dried in a shady place at 25–30 °C for seven days, and then stored at −18 °C until further utilization. Before extraction, all the dried plant samples (aerial parts, i.e., leafy stems and corymb inflorescences in full bloom) were cut in a laboratory blender and sieved through a Vibratory Sieve Shaker Analysette 3 (Fritsch GmbH, Idar-Oberstein, Germany), and had a particle size of <400 um and a moisture content of <5% *w/w*, which were measured gravimetrically.

### 3.2. Reagents and Solvents

For the HPLC and LC-MS analyses, the solvents HPLC grade acetonitrile (ACN) ≥ 99.9% (Honeywell Riedel-de Haën GmbH, Seelze, Germany), LC-MS grade methanol (MeOH) ≥ 99.9% (Merck, Darmstadt, Germany), formic acid (HCOOH) 98% (PENTA s.r.o., Prague, Czech Republic), and purified water supplied by a reversed osmosis water purification system (Human^®^ Corporation, Seoul, Republic of Korea) were used. Kaempferol 3-β-D-glucopyranoside (astragalin) ≥ 97.0%, chlorogenic acid ≥ 95.0%, and neochlorogenic acid ≥ 98.0% reference standards were purchased from Sigma Aldrich (St. Louis, MO, USA). For the total polyphenol content (TPC assay), gallic acid monohydrate > 98% and sodium carbonate anhydrous > 99% were used (PENTA s.r.o., Prague, Czech Republic), whereas distilled water was used for the sample dilutions and extractions. For the hydromethanolic extractions, MeOH for HPLC ≥ 99.9% was used (Honeywell Riedel-de Haën GmbH, Seelze, Germany).

### 3.3. Extraction and Separation Methods

#### 3.3.1. Preparation of Methanol Extracts for LC-HRMS/MS Analyses

The procedure for preparing the methanol total extracts of the 11 Greek *Helichrysum* taxa for LC-HRMS analysis has been described analytically in our previous studies [28]. In brief, 230 ± 10 mg of plant material was placed in a glass vial with 5 mL LC-MS-grade MeOH, extracted in an ultrasonic bath for 15 min at a temperature range of 25–40 °C, and filtered through a 0.45 μm PTFE syringe filter (Membrane Solutions, Kent, WA, USA). The extraction was repeated with fresh solvent under the same conditions. The filtered extracts were combined and dried using a rotary evaporator (Laborota 4000 efficient, Heidolph Instruments GmbH and Co. KG, Schwabach, Germany) in a water bath (<40 °C) and under high vacuum until a constant weight was reached. The dried extracts were reconstituted with LC-MS-grade MeOH and ultrapure water to a final concentration of 400 μg/mL before undertaking LC-MS analysis.

#### 3.3.2. Preparation of Kinetic Maceration Extracts for Total Polyphenols and HPLC Analyses

Kinetic maceration (stirring, STE) of the HA plant material was performed in DURAN^®^ bottles at ambient temperatures, using MSH-20A magnetic stirring plates (DURAN, Witeg, Wertheim, Germany). Approximately 1 g of plant material was mixed with 10 mL (Drug-Solvent Ratio, DSR = 1:10) of the appropriate solvent—MeOH, MeOH:H_2_O (9:1 *v/v*), MeOH:H_2_O (7:3 *v/v*), or H_2_O—under stirring at 500 rpm for 60 min. Solid–liquid separation was achieved by centrifugation (Neya Centrifuges, Carpi, Italy) at 4500 rpm for 5 min. The same plant material was re-extracted with 10 mL of fresh solvent, and after centrifugation, the liquid extracts were combined.

Kinetic maceration of the other *Helichrysum* taxa plant materials was performed as described above, in MeOH:H_2_O (7:3 *v/v*), and treated similarly.

#### 3.3.3. Preparation of PLE Extracts for Total Polyphenols and HPLC Analyses

Pressurized liquid extraction (PLE) of the HA plant material was conducted on a PLE™ Extraction System (Fluid Management Systems, Inc., Billerica, MA, USA). In brief, approximately 4 g of plant material was placed in a 40 mL stainless steel extraction cell capped with two filtration end fittings to perform static extraction. After sample loading, solvent pumping (40 mL of distilled H_2_O, DSR = 1:10), cell pressurization, and heating occurred for predetermined time intervals. Each extraction cycle was completed under a set method, as follows: filling column time, 1.3 min; pressurization, 0.5 min; preheating (175 °C) time, 5 min; extraction (175 °C) time, 15 min; cooling (<60 °C), 10 min; depressurization time, 0.02 min; nitrogen flush, 3 min. The extraction of the same plant material was repeated with fresh solvent (two extraction cycles in total), and the two liquid extracts were combined. Solid–liquid separation was facilitated by centrifugation at 4500 rpm for 5 min.

All the liquid extracts were kept at −40 °C until the day of analysis.

### 3.4. Analytical Methods

#### 3.4.1. UPLC-HRMS/MS Profiling

The procedure for analyzing the samples of MeOH extracts of the 11 Greek *Helichrysum* taxa by LC-HRMS/MS has been described analytically in previous studies [28]. In brief, the UPLC-HRMS/MS analysis for the untargeted secondary metabolite profiling was conducted using an ACQUITY UPLC H-Class system (Waters, Milford, MA, USA), interfaced to an OrbiTrap Velos Pro Hybrid mass spectrometer (Thermo Scientific, Waltham, MA, USA) with a HESI (Heated Electrospray Ionization) source. Chromatographic separation was performed with a UPLC^®^—Column Acquity BEH C18 (50 mm × 1.0 mm; 1.7 μm), involving a gradient of water with 0.1% (*v*/*v*) formic acid (solvent A) and acetonitrile (solvent B), at a flow rate of 400 μL/min. The column and autosampler were thermostated at 40 °C and 7 °C, respectively, and the sample injection volume was 10 μL. Elution was achieved under the following gradient programme: initial conditions: 0 min, (B) 5%; 0–1 min, (B) increase 5–15%; 1–15 min, (B) increase 15–100%; 15–17 min, (B) isocratic 100%; 17–17.5 min, (B) decrease 100–95%; and 17.5–20.0 min, (B) equilibration to initial conditions 5%. The total run time was 20 min. Mass spectra were obtained using a *m/z* range of 113–1000 Da for full scan acquisition, at a resolving power of 30,000 (full width half maximum, FWHM, at 500 *m/z*), with a scanning rate of 1 micro scan/sec, and a mass error below 5 ppm. Sheath and auxiliary gas were appointed at 45 and 15 arbitrary units, respectively. For profiling of all the samples, the negative ESI mode was selected. HRMS/MS spectra were recorded using data-dependent acquisition (FS-dd-MS2), with a collision energy of 35.0% (q = 0.25) and a resolving power of 30,000 at 500 *m/z*. The capillary and source heater temperatures were set at 350 °C, the source voltage at 2.7 kV, and the S-lens RF level at 45%. For positive ESI, the same parameters were used, except for the source voltage (3.6 kV) and the S-lens RF level (60%). Xcalibur^™^ 2.0.7 (Thermo Scientific, Waltham, MA, USA) software was used for data acquisition and processing. Peak selection was based on their intensities (>1 × 10^5^ of the noise level), and the metabolite tentative identification was accomplished based on the signal of the main molecular ion (in positive or negative ionization), the Elemental Composition (EC), the Ring Double Bond equivalent (RDBeq), and by comparing the fragmentation profile of each peak with data from the literature.

#### 3.4.2. RP-HPLC-DAD Analysis

RP-HPLC-DAD analyses were performed on a Shimadzu Prominence LC system (Shimadzu Europa GmbH, Duisburg, Germany), consisting of a CBM-20Alite system controller, a DGU-20A5 degasser, an LC-20AD pump, a SIL-20AC autosampler, a CTO-20AC oven, and an SPD-M20A diode array detector. Chromatographic separation was carried out on a Phenomenex Luna 5 µm, C18(2), 100 Å, 250 mm × 4.6 mm, Ea, LC column (Phenomenex, Torrance, CA, USA). The oven temperature was set at 40 °C. For the chromatographic elution, a mobile phase consisting of 0.5% HCOOH in water (Solvent A) and 0.5% HCOOH in acetonitrile (Solvent B) was used. Elution was achieved by applying the following low-pressure method: 0–40 min, (B) increase 0–24%; 40–50 min, (B) increase 24–30%; 50–60 min, (B) increase 30–42%; 60–70 min, (B) isocratic 42%; 70–75 min, (B) increase 42–100%; 75–80 min, (B) isocratic 100%; and 80–85 min, (B) decrease 100–0%, at a flow rate of 1 mL/min and 20 μL injection volume. Before analysis, all liquid extracts were clarified by centrifugation at 10,000 rpm for 10 min, and diluted with ultrapure water (1:1). Elution was monitored between 200 and 800 nm. All chromatographic data were processed utilizing LC Solution Version 1.22 SP1 software (Shimadzu Corporation, Kyoto, Japan) and OpenChrom^®^ Lablicate Edition (McLafferty) version 1.5.0. Kaempferol-3-*O*-glucoside was quantified by peak integration at 265 nm and in a specific retention time window. A standard calibration curve was used (y = 50,916.84x − 42,398.82, *R*^2^ = 0.9962, concentration range 0–50 ppm). The results were expressed as mg/L (for liquid extract) and mg/g (dry plant material). Chlorogenic and neochlorogenic acids were quantified in the same manner by peak integration at 324 nm. The following standard calibration curves were used: y = 50,320.40x − 23,038.36, *R*^2^ = 0.994 for chlorogenic acid; and y = 28,213.52x + 551.72, *R*^2^ = 0.999 for neochlorogenic acid, at a concentration range of 0–50 ppm.

#### 3.4.3. UV-Vis Analysis

UV-vis analysis was performed on a UV-1700 Shimadzu spectrophotometer (Shimadzu Europa GmbH, Duisburg, Germany). After extraction, each liquid extract was clarified by centrifugation at 10,000 rpm for 10 min, diluted with ultrapure water (1:100), and scanned in the range of 190–700 nm for the identification of λ_max_ absorbance.

#### 3.4.4. Total Polyphenol Content Determination

For the determination of total polyphenol content (TPC assay), all liquid extracts were clarified by centrifugation at 10,000 rpm for 10 min, and diluted appropriately with ultrapure water. Then, 100 μL of the diluted extract was combined with 100 μL of the Folin–Ciocalteu reagent in a 1.5 mL Eppendorf tube. After a two-minute reaction time, 800 μL of 5% *w/v* sodium carbonate solution was introduced, followed by a 20 min incubation at 40 °C. Absorbance measurements were performed at 740 nm in a UV-1700 Shimadzu spectrophotometer. A gallic acid calibration curve was constructed in the linear range of concentrations 10–100 mg/L, with the equation y = 0.0119x + 0.0114, *R*^2^ = 0.9995. The extraction yield of total polyphenols (*Y*_TP_) was calculated according to Equation (1), and was expressed as mg gallic acid equivalents (GAE) per g of dry weight (dw) of plant material:(1)$$\mathrm{TPC}\;(\mathrm{mg}\;\mathrm{GAE}/g\;\mathrm{dw})=\frac{C_{\mathrm{TP}}\;\times \;V}{w}$$
where *C*_TP_ denotes the total polyphenol concentration (mg GAE/L), *V* is the volume of the extraction medium (L), and *w* is the dry weight of the plant material (g).

### 3.5. Statistical Analysis

The results were reported as mean values, with standard deviations (SD) derived from triplicate analyses. To evaluate the statistical significance of differences among the means, the Tukey–Kramer test was applied at a threshold of *p* < 0.05. All related statistical analyses were carried out using JMP^®^ Pro 16 software (SAS Institute, Cary, NC, USA).

## 4. Conclusions

This study contributes to the exploration of the non-volatile phytochemical profile of the 11 *Helichrysum* taxa found in Greece, following a dereplication study on their respective methanol extracts, by primarily identifying metabolites known to be present in these members of the genus *Helichrysum*; among them, novel profiles are revealed for five Greek endemic *Helichrysum* species which are Endangered (*H. amorginum*, *H. taenari*) or Critically Endangered (*H. doerfleri*, *H. heldreichii*, *H. sibthorpii*). Furthermore, the TPC of the hydromethanolic liquid extracts from the studied taxa is comparatively assessed, revealing significant differences among most taxa, showing similarity to reported profiles of *H. arenarium*. A special focus was placed on the cultivated genotype of *H. amorginum* (HA) derived directly from wild-growing plants, revealing a comparable total polyphenol yield for the PLE setup to that of a hydromethanolic mixture. Regarding the neochlorogenic acid, chlorogenic acid, and astragalin contents in the studied HA genotype, PLE extraction resulted in a higher content of neochlorogenic acid compared to water–methanol and water extraction. On the other hand, no increase in the chlorogenic acid content was observed. In addition, PLE extraction resulted in a higher astragalin content compared to maceration in water. Overall, the present findings offer a new perspective for more in-depth research on these *Helichrysum* taxa, highlighting their potential future industrial applications in food supplements, cosmetics, and fine chemical production in the frame of sustainable exploitation strategies.

## Figures and Tables

**Figure 1 plants-14-00229-f001:**
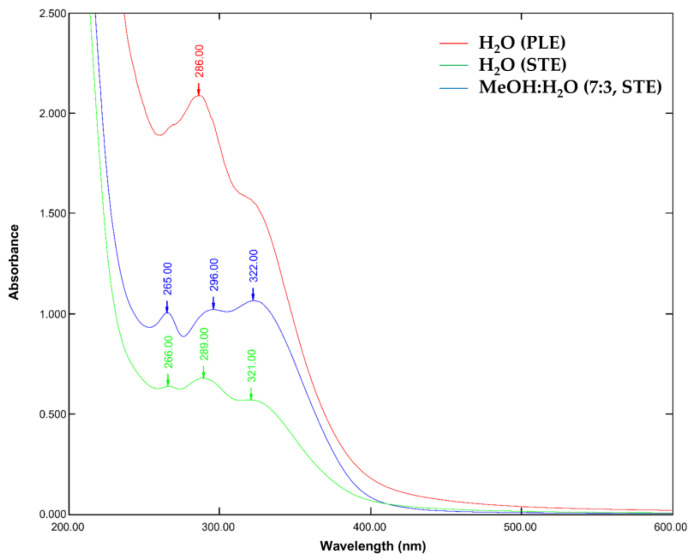
UV-vis absorption spectra of the *Helichrysum amorginum* (HA) diluted extracts H_2_O (PLE), H_2_O (STE), and MeOH:H_2_O (7:3, STE).

**Table 1 plants-14-00229-t001:** LC-HRMS/MS ^1^ tentative identification of secondary metabolites in MeOH extracts of wild plant material of 11 Greek *Helichrysum* taxa.

Taxon’s Scientific Name (Abbreviation)	Compounds
*Helichrysum amorginum* Boiss. and Orph. (HA) ^2^	**Acids and derivatives:** Gluconic acid, Quinic acid, Malic acid, 2-Isopropyl malic acid, Syringic acid 4-*O*-β-D glucopyranoside, Caffeic acid, Caffeoyl glycerol, Caffeoyl quinic acid isomers, Dicaffeoylquinic acid isomers, Tricaffeoyl hexaric acid, Pinellic acid, 3,5-Dihydroxy-hexadecanoic acid, Trihydroxy octadecenoic acid, Dihydroxy octadecadienoic acid, Octadecanedioic acid, Helenylonic acid (9-hydroxy-10E-octadecen-12-ynoic acid), Hydroxy octadecadienoic acid **Flavonoids:** Quercetin, Quercetin glycoside, Apigenin, Apigenin-7-*O*-glucoside, Eriodictyol, Kaempferol, Kaempferol 3-*O*-acetylhexoside, Kaempferol-3-*O*-malonyl-hexoside, Kaempferol 3-*O*-acetylhexoside, Kaempferol-3-*O*-glucoside, Kaempferol dihexoside, Kaempferol-3-*O*-sophoroside-7-*O*-glucoside, Kaempferol-3-*O*-[6″-*O*-(*trans*-*p*-coumaroyl)-3″-*O*-acetyl]-β-D-glucopyranoside, Tiliroside, Tribuloside, Naringenin, Isorhamnetin, 5,3′,4′-trihydroxy-6,7,8-trimethoxyflavone **Pyrones and derivatives:** Cycloarzanol, Italipyrone, Heliarzanol, Arenol, Arzanol, 3-Methylarzanol, Unknown pyrone derivatives **Others:** Araneophthalide, Acronyline, Tetrahydrocurcumin, Oleylbitalin A, Unknown prenylated polyphenols, Unknown prenyl phloroglucinol derivatives
*Helichrysum doerfleri* Rech. f. (HD)	**Acids and derivatives:** Quinic acid, Malic acid, Caffeic acid, Caffeoyl hexoside, Caffeoylquinic acid isomers, Dicaffeoylquinic acid isomers, 5-*O*-Feruloylquinic acid, 3,5-Dihydroxy-hexadecanoic acid, C17 Hydroxy fatty acid, Linolenic acid, C18 Trihydroxy fatty acid, C18 Hydroxy fatty acid **Flavonoids:** Quercetin, Quercetin glycosides, Quercetin-7-*O*-(caffeoyl)-hexoside, Kaempferol-3-*O*-glucoside, Kaempferol-*O*-acetylhexoside, Kaempferol dihexoside, Kaempferol-3-*O*-sophoroside-7-*O*-glucoside, Kaempferol-3-*O*-[6″-*O*-(*trans*-*p*-coumaroyl)-3″-*O*-acetyl]-β-D-glucopyranoside, Tiliroside, unknown methylated flavonoids **Others:** unknown prenylated polyphenols, unknown terpenes
*Helichrysum heldreichii* Boiss. (HH)	**Acids and derivatives:** Glucoheptonic acid, Malic acid, Protocatechuic acid *O*-hexoside, Caffeic acid, Caffeoylquinic acid isomers, Dicaffeoylquinic acid isomers, Dicaffeoylquinic acid methyl ester, 5-*O*-Feruloylquinic acid, C18 Trihydroxy fatty acids **Flavonoids:** Quercetin, Quercetagetin-3,7-di-*O*-hexoside, Quercetin glycosides, Quercetin coumaroylglucoside, Quercetin-7-*O*-(caffeoyl)-hexoside, Quercetin coumaroylglucoside analogue, Kaempferol, Kaempferol dihexoside, Unknown methylated flavonoids **Others:** Pinobanskin, Dilignol analogues, Unknown triterpenes, Aesculin, Aesculetin
*Helichrysum luteoalbum* (L.) Rchb. (HL)	**Acids and derivatives:** Quinic acid, Malic acid, Succinic acid, Protocatechuic acid-*O*-hexoside, 2,4-Dihydroxybenzoic acid, Caffeic acid, Caffeoyl hexoside, Caffeoylquinic acid isomers, Dicaffeoylquinic acid isomers, 5-*O*-Feruloylquinic acid, C18 Trihydroxy fatty acid **Flavonoids:** Apigenin, Apigenin-6-*C*-glucoside, Apigenin caffeoylhexoside isomers, Kaempferol-3-*O*-glucoside, Kaempferol-7-*O*-hexoside, Kaempferol dihexoside, Luteolin, Quercetin glycosides, Unknown methylated flavonoids **Pyrones and derivatives:** Auricepyron analogue **Others:** Unknown diterpene, Oxygenated aliphatic hydrocarbons, Lupulone
*Helichrysum orientale* (L.) Gaertn. (HO)	**Acids and derivatives:** Malic acid, Syringic acid 4-*O*-β-D-glucopyranoside, Caffeic acid, Caffeoyl hexoside, Caffeoylquinic acid isomers, Dicaffeoylquinic acid isomers, 5-*O*-Feruloylquinic acid, C17 Hydroxy fatty acid, C18 Trihydroxy fatty acid **Flavonoids:** Naringenin, Naringenin-*O*-hexosides, Kaempferol, Kaempferol-3-*O*-glucoside, Tiliroside, unknown methylated flavonoids **Others:** unknown terpenes, hexose
*Helichrysum plicatum* DC. (HP)	**Acids and derivatives:** Glucoheptonic acid, Quinic acid, Malic acid, Protocatechuic acid-*O*-hexoside, Caffeic acid, Caffeoyl hexoside, Caffeoylquinic acid isomers, Dicaffeoylquinic acid isomers, 5-*O*-Feruloylquinic acid **Flavonoids:** Isovitexin 2″-*O*-β-D-glucoside, Apigenin, Apigenin-6-C-glucoside, Naringenin-*O*-hexosides, 5,6,4′-Trihydroxy-3′-methoxyflavone-7-*O*-β-glucoside, Luteolin, Dihydrokaempferol, Unknown methylated flavonoids **Pyrones and derivatives:** Helicerastripyrone
*Helichrysum sibthorpii* Rouy (HS)	**Acids and derivatives:** Quinic acid, Malic acid, 1-*O*-Protocatechuyl-β-D-xylopyranose, Caffeic acid, Caffeoyl hexoside, Caffeoylquinic acid isomers, Dicaffeoylquinic acid isomers, 5-*O*-Feruloylquinic acid, Oxododecanedioic acid, C18 Trihydroxy fatty acids **Flavonoids:** Quercetin glycosides, Tiliroside, unknown methylated flavonoids **Pyrones and derivatives:** Arzanol, 3-Methylarzanol, 3-Methylarzanol isomer, Plicatipyrone analogues, Auricepyron analogues, unknown pyrone derivatives **Others:** unknown prenylated derivatives, unknown phloroglucinol, Araneophthalide, Lupulone, Coumarin analogues
*Helichrysum taenari* Rothm. (HT)	**Acids and derivatives:** Quinic acid, Malic acid, Caffeic acid, Caffeoyl hexoside, Caffeoyl glycerol, Caffeoylquinic acid isomers, 5-*O*-Feruloylquinic acid, Dicaffeoylquinic acid isomers, C17 Hydroxy fatty acids, C18 Trihydroxy fatty acid **Flavonoids:** Apigenin-6-*C*-glucoside, Apigenin, Kaempferol diglycosides, Kaempferol-3-*O*-glucoside, Tiliroside, Tribuloside, Luteolin glycosides, Luteolin, Quercetin glycosides, Pinostrobin, unknown methylated flavonoids **Pyrones and derivatives:** Arzanol analogue **Others:** unknown terpenes
*Helichrysum italicum* (Roth) G.Don subsp. *italicum* (HII)	**Acids and derivatives:** Quinic acid, Malic acid, Protocatechuic acid-*O*-hexoside, 2,4-Dihydroxybenzoic acid, Caffeic acid, Caffeoyl hexoside, Caffeoylquinic acid isomers, Dicaffeoylquinic acid isomers, Malonyl-dicaffeoylquinic acid, 4-Feruloyl-5-caffeoylquinic acid, Hydroxycinnamic acid derivatives, C17 hydroxy fatty acid, C18 Trihydroxy fatty acid isomers, unknown triterpenic acids **Flavonoids:** Delphinidin-3-glycoside, Eriodictyol-*O*-hexoside, Kaempferol dihexoside, Myricetin-3-*O*-glucoside, Myricetin malonylhexoside, Quercetin glycosides, Quercetin **Others:** Dilignol analogue, unknown terpenes, Aesculin
*Helichrysum italicum* (Roth) G.Don subsp. *microphyllum* (Willd.) Nyman (HIM)	**Acids and derivatives:** Quinic acid, Malic acid, Protocatechuic acid-*O*-hexoside, 2-Isopropyl malic acid, Caffeic acid, Caffeoylquinic acid isomers, Dicaffeoylquinic acid isomers, Hydroxycinnamic acid derivative, Hydroxyjasmonate **Flavonoids:** Gnaphaliol-*O*-β-D-glucopyranoside, Quercetin glycosides, Quercetin coumaroylglucoside analogue, Tiliroside, Luteolin, Quercetin, Isorhamnetin, Naringenin, Pinobanskin, Pinocembrin, Homoeriodictyol, Trihydroxy-methoxyflavone, 5,7-Dihydroxy-3-methoxyflavone **Pyrones and derivatives:** 6-ethyl-4-hydroxy-5-methyl-3-(3-oxopentyl)-2H-pyran-2-one, 7-(2,3-dihydroxy-3-methylbutoxy)-5-hydroxy-6-methoxy-2H-1-benzopyran-2-one, Micropyrone, Plicatipyrone, Plicatipyrone analogues, Arenol, Arzanol, 3-Methylarzanol, Heterodimer pyrone-phloroglucinol, Heliarzanol analogue, Auricepyron analogue, unknown pyrone derivatives, Italipyrone **Others:** Coumarin analogue, Araneophthalide, 6-Demethylacronylin
*Helichrysum stoechas* (L.) Moench subsp. *barrelieri* (Ten.) Nyman (HSB)	**Acids and derivatives:** Quinic acid, Malic acid, Protocatechuic acid-*O*-hexoside, Protocatechuic acid, Caffeic acid, Caffeoyl hexoside, Caffeoylquinic acid isomers, Dicaffeoylquinic acid isomers, 5-*O*-Feruloylquinic acid, Malonyl-dicaffeoylquinic acid, Feruloyl-caffeoyl-quinic acid, C17 Hydroxy fatty acids, C18 Trihydroxy fatty acids, unknown triterpenic acids **Flavonoids:** Myricetin-3-*O*-glucoside, Quercetin, Quercetin glycosides, Quercetin-7-*O*-(caffeoyl)-hexoside, Kaempferol dihexoside, Tiliroside, Pinocembrin, unknown methylated flavonoids **Others:** Dilignol analogue, unknown terpenes

^1^ LC-HRMS/MS chromatographic and spectrometric data are presented in the Appendix A (HA, Appendix A; HD, Appendix A; HH, Appendix A; HL, Appendix A; HO, Appendix A; HP, Appendix A; HS, Appendix A; HT, Appendix A, HII, Appendix A; HIM, Appendix A; HSB, Appendix A). ^2^ Cultivated genotypes directly derived from a wild-growing population.

**Table 2 plants-14-00229-t002:** Total polyphenol contents in the 11 Greek *Helichrysum* taxa studied (dry plant material and liquid extracts). For abbreviations of taxa, see Table 1.

*Helichrysum* Taxa	*C*_TP_ (mg GAE/L ± SD)	Y_TP_ (mg GAE/g dw) ± SD ^1^
HA	2056.51 ± 37.83	41.02 ± 0.75 ^g^
HD	1699.71 ± 55.92	32.86 ± 1.08 ^h^
HH	1758.57 ± 10.09	34.27 ± 0.20 ^h^
HL	1355.11 ± 10.74	25.34 ± 0.20 ^j^
HO	1577.23 ± 15.46	29.08 ± 0.29 ^i^
HP	5106.33 ± 42.73	98.85 ± 0.83 ^b^
HS	2316.75 ± 79.11	47.72 ± 1.63 ^f^
HT	3386.57 ± 30.30	64.75 ± 0.58 ^d^
HII	2439.65 ± 19.19	57.57 ± 0.45 ^e^
HIM	5683.70 ± 47.83	107.97 ± 0.91 ^a^
HSB	3855.48 ± 48.50	76.60 ± 0.96 ^c^

^1^ Mean values, n = 3; different superscript letters a–j denote statistically significant differences between mean values (*p* < 0.05).

**Table 3 plants-14-00229-t003:** Total polyphenol contents in *Helichrysum amorginum* (dry plant material and liquid extracts), according to different extraction conditions.

Solvent/Method/Conditions	Static Dielectric Constant (ε)	Y_TP_ (mg GAE/g dw) ± SD ^1^ (*C*_TP_, mg GAE/L ± SD)
MeOH/STE ^2^/24 °C	≅33 (25 °C)	20.32 ± 0.26 ^d^ (1017.34 ± 12.71)
MeOH:H_2_O (9:1)/STE/21–22 °C	36.1 (20 °C)	27.04 ± 0.95 ^c^ (1366.88 ± 48.06)
MeOH:H_2_O (7:3)/STE/21–22 °C	42.54 (20 °C)	41.02 ± 0.75 ^a^ (2056.51 ± 37.83)
H_2_O/PLE/175 °C/1450 psi (≅10 MPa)	≅39.5	37.6 ± 0.16 ^b^ (1886.52 ± 7.79)
H_2_O/STE/19 °C	≅80 (20 °C)	21.99 ± 0.94 ^d^ (1100.64 ± 47.16)

^1^ Mean values, n = 3; different superscript letters a–d denote statistical significance in mean values (*p* < 0.05). ^2^ Kinetic maceration (i.e., stirring, STE).

**Table 4 plants-14-00229-t004:** *λ*_max_ absorbances of the *Helichrysum amorginum* (HA) diluted extracts H_2_O (PLE), H_2_O (STE), and MeOH:H_2_O (7:3, STE).

Solvent	Method	*λ* _max_
H_2_O	PLE	266, 286, 321
H_2_O	STE	266, 289, 321
MeOH:H_2_O (7:3)	STE	265, 296, 322

**Table 5 plants-14-00229-t005:** Neochlorogenic acid (NCA), chlorogenic acid (CA), and astragalin (AST) concentration in the studied *Helichrysum amorginum* (HA) genotype (dry plant material and liquid extract).

Metabolite Concentration	H_2_O (PLE) ^1^	H_2_O (STE) ^1^	MeOH:H_2_O (7:3, STE) ^1^
NCA mg/L extract (SD)	130.91 (1.28) ^a^	19.83 (1.32) ^c^	24.41 (0.31) ^b^
CA mg/L extract (SD)	115.64 (0.66) ^c^	134.66 (0.83) ^b^	188.67 (0.41) ^a^
AST mg/L extract (SD)	206.57 (3.32) ^b^	143.60 (1.57) ^b^	694.36 (116.06) ^a^
NCA mg/g plant material (SD)	2.61 (0.03) ^a^	0.40 (0.03) ^c^	0.49 (0.01) ^b^
CA mg/g plant material (SD)	2.30 (0.01) ^c^	2.69 (0.02) ^b^	3.76 (0.01) ^a^
AST mg/g plant material (SD)	4.12 (0.07) ^b^	2.87 (0.03) ^b^	13.85 (2.32) ^a^

^1^ Mean values, *n* = 3; different superscript letters ^a^–^c^ in each row indicate statistical significance in mean values (*p* < 0.05).

## Data Availability

The original data presented in this study are included in the article and its Appendix A. Further inquiries can be directed to the corresponding authors.

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
