# Peer review of "A Comparative Phytochemical Investigation of the Greek Members of the Genus Helichrysum Mill., with Emphasis on the Local Endemic Helichrysum amorginum Boiss and Orph"

_plants, 2025, doi:10.3390/plants14020229_

Round 1
Reviewer 1 Report
Comments and Suggestions for Authors
I read with interest the paper presented by Samanidis et al. who evaluated the non-volatile phytochemical profile of 11 Greek Helichrysum taxa. They compared the total polyphenolic content of these endangered and critically endangered species. In addition, they quantified the major metabolites of Helichrysum amorginum using pressurized liquid extraction, UV-Vis and HPLC analysis. On the other hand, they analyzed by HPLC the content of neochlorogenic acid, chlorogenic acid and astragalin in genotypes of H. amorginum. The authors should clarify some aspects before the possible acceptance of this work in MDPI-Plants.
1. The authors highlight that Helichrysum species could be sustainable sources for potential industrial applications in the production of food supplements, cosmetics and fine chemical production, using more environmentally friendly extraction techniques. However, the plant material used is from endangered or critically endangered species. The authors do not explicitly state how these species could be a sustainable source given their conservation status. What alternatives are available to ensure the sustainable use of endangered plant species for these industrial applications?
2. The authors should explain why the pressurized liquid extraction (PLE) method is an environmentally friendly technique. What are the advantages and disadvantages of this technique?
3. Authors must also justify the use of maceration as an extraction method.
4. It is suggested that the authors include the common names of the Helichrysum species used.
5. To improve understanding, I suggest that section 3.4 corresponding to analytical methods be subdivided by separating the experimental strategies of chromatographic methods, UV-vis analysis and the determination of total phenols.
6. It is recommended to provide the extraction yields of the different preparation techniques of the extracts of Helichrysum species.
7. The chromatographic analysis concentrated on identifying neochlorogenic acid, chlorogenic acid, and astragalin. Nevertheless, the rationale for selecting these specific secondary metabolites over other reported compounds within the genus remains ambiguous.
8. It is recommended to analyze the metabolite contents in comparison with those from other plants within the same genus to gain a more comprehensive understanding of the significance of the findings.
9. It is recommended that the authors examine the constraints of the analytical techniques employed in the study and specify if additional research is necessary to explore the content of secondary metabolites in Greek Helichrysum taxa.
10. The conclusions must be rewritten to eliminate methodological aspects that were already described in the methods.
Author Response
I read with interest the paper presented by Samanidis et al. who evaluated the non-volatile phytochemical profile of 11 Greek Helichrysum taxa. They compared the total polyphenolic content of these endangered and critically endangered species. In addition, they quantified the major metabolites of Helichrysum amorginum using pressurized liquid extraction, UV-Vis and HPLC analysis. On the other hand, they analyzed by HPLC the content of neochlorogenic acid, chlorogenic acid and astragalin in genotypes of H. amorginum. The authors should clarify some aspects before the possible acceptance of this work in MDPI-Plants.
- The authors highlight that Helichrysumspecies could be sustainable sources for potential industrial applications in the production of food supplements, cosmetics and fine chemical production, using more environmentally friendly extraction techniques. However, the plant material used is from endangered or critically endangered species. The authors do not explicitly state how these species could be a sustainable source given their conservation status. What alternatives are available to ensure the sustainable use of endangered plant species for these industrial applications?
We would like to thank the reviewer for this comment aimed at the manuscript’s improvement. Following the suggestion made by the reviewer, we have rephrased parts of the introduction and results and discussion sections, stating explicitly in the revised version of the manuscript that such range-restricted and threatened Greek endemic species can only be in depth investigated or used industrially when they are sustainably managed and genotyped of wild origin are established in ex-situ cultivation (see track changes, respectively). This is also clearly stated in the aim of the study (see track changes).
- The authors should explain why the pressurized liquid extraction (PLE) method is an environmentally friendly technique. What are the advantages and disadvantages of this technique?
Pressurized Liquid Extraction (PLE) is widely regarded as an environmentally friendly extraction method, particularly when compared to traditional techniques like Soxhlet extraction or maceration. The following points highlight why PLE is considered eco-friendly:
- Reduced Solvent Usage: PLE requires significantly less solvent than traditional methods, lowering the environmental impact associated with solvent production, handling, and disposal.
- Green Solvents Options: PLE commonly employs "green" solvents such as water, and ethanol, which are non-toxic, renewable, and biodegradable, offering a sustainable alternative to petroleum-based solvents.
- Energy Efficiency: Operating at moderate temperatures (50–200°C) and elevated pressures, PLE consumes less energy than techniques that demand prolonged heating or cooling.
- Waste Minimization: The method is highly efficient, achieving high extraction yields while reducing the volume of plant material and solvents required, thus generating less waste.
- Closed-Loop Systems: Most PLE systems are designed as closed loops, enabling the recycling and reuse of solvents and materials, which minimizes emissions and environmental exposure.
- Versatility: PLE is capable of extracting a wide range of compounds, reducing the need for multiple separate extraction processes, and thereby lowering the overall environmental footprint.
- Limitations: Despite its eco-friendly advantages, PLE’s sustainability can vary depending on:
- The choice of solvent (e.g., petroleum-based solvents are less sustainable).
- The energy source powering the equipment (renewable energy sources enhance its environmental benefits).
- What residual solvents and extracted materials are managed and disposed of.
Nonetheless, as no energy measurements and LCA study were conducted in the present study regarding the environmental performance of PLE vs maceration (stirring) extraction methods the word “environmental” was deleted from the manuscript.
- Authors must also justify the use of maceration as an extraction method.
Maceration (referred to as kinetic maceration or stirring, STE, in the manuscript) is a dynamic extraction method widely used for the extraction of Medicinal and Aromatic Plants (according to United Nations Industrial Development Organization and the International Centre for Science and High Technology), for the preparation of Herbal Teas (according to European Pharmacopoeia and European Medicines Agency), and for the preparation of Herbal Drug Extracts such as tinctures (according to European Pharmacopoeia). This traditional method was selected as a reference extraction method for quantitative and qualitative comparisons with other advanced extraction techniques like Pressurized Liquid Extraction. Moreover, it was selected as a reference extraction method instead of the Soxhlet extraction avoiding the prolonged heating of the extract at the solvent’s boiling point which may lead to extract deterioration. Additionally, it was selected because it is considered a non-exhaustive extraction method with often unfavourable performance in terms of extraction yield and environmental impact.
- It is suggested that the authors include the common names of the Helichrysumspecies used.
We would like to thank the reviewer for this comment. Although we would like to be able to respond positively to the suggestion made, most unfortunately, the Helichrysum taxa in Greece do not have official common names. Commonly, their genus name of Greek origin is often used by people universally, only discriminating geographically different species, e.g., helichryso (Greek version of Latin Helichrysum) of Amorgos (amorgino helichryso) or helichryso of Cape Taenaron (Helichrysum taenari) or referring to their original botanical descriptions (e.g., helichryso of Sibthorp for H. sibthorpii or helichryso of Heldreich for H. helderichii). However, to accommodate the reviewer’s suggestion, the first case is referred to in the revised version of the manuscript regarding the herein focal H. amorginum which is also called ‘stathouri’ locally in Amorgos Island (see track changes in the revised introduction section).
- To improve understanding, I suggest that section 3.4 corresponding to analytical methods be subdivided by separating the experimental strategies of chromatographic methods, UV-vis analysis and the determination of total phenols.
As suggested by the reviewer, different subsubsections have been added in subsection 3.4 to improve understanding in the revised version of the manuscript (see track changes).
- It is recommended to provide the extraction yields of the different preparation techniques of the extracts of Helichrysumspecies.
Extraction yields of the different preparation techniques measured in % dry extract yield per mass of dry weight of plant material were nοt determined to avoid liquid extract losses during the liquids transferring and thermal stress and during concentration before drying. Instead, the yield measurement of the preparation techniques was conducted on total polyphenols content and was expressed as concentration (mg GAE/L) of the liquid extract and as mg GAE/g dry weight (dw) of plant material. Also, the yield measurement of the preparation techniques was conducted on the content of the metabolites CA, NCA, and astragalin and was expressed as concentration (mg/L) of the liquid extract and as mg/g dry weight (dw) of plant material.
- The chromatographic analysis concentrated on identifying neochlorogenic acid, chlorogenic acid, and astragalin. Nevertheless, the rationale for selecting these specific secondary metabolites over other reported compounds within the genus remains ambiguous.
The members of the genus Helichrysum are renowned for their content of flavonoids and caffeoylquinic acid derivatives. Astragalin was selected as the major representative metabolite in the category of flavonoids in Helichrysum amorginum. Chlorogenic and neochlorogenic acids were selected as representatives of the caffeoylquinic acids in the same species. All metabolites were quantified using available reference standards. These secondary metabolites were also chosen because of their known bioactivity as research studies imply.
- It is recommended to analyze the metabolite contents in comparison with those from other plants within the same genus to gain a more comprehensive understanding of the significance of the findings.
We would like to thank the reviewer for this comment. In this study, the contents of CA, NCA, and AST in H. amorginum were measured as concentration (mg/L of liquid extract) and as content expressed in mg/g of dry weight (dw) plant material. Thereby attention was paid to comparing the CA, NCA, and AST metabolites contents of H. amorginum with other Mediterranean species under the same units, i.e., mg/g or mg/kg. For that reason, the content of CA, and NCA in H. amorginum was compared with the content in H. italicum (expressed in mg/g dw plant material) and the content of AST in H. amorginum was compared with the content in H. italicum and H. stoechas subsp. barrelieri (expressed in mg/kg dw plant material). Limited citations were available for comparison using the same units. Usually, the metabolite content is expressed in w/w dry weight of extract and not in w/w dry weight of plant material.
- It is recommended that the authors examine the constraints of the analytical techniques employed in the study and specify if additional research is necessary to explore the content of secondary metabolites in Greek Helichrysumtaxa.
We would like to thank the reviewer for this comment. In this study, the UPLC-HRMS/MS technique offered a first (tentative) identification of the metabolites in the MeOH extracts of the Greek Helichrysum taxa (species and subspecies). The UV-vis technique aimed to justify the selection of λmax used in the HPLC-DAD analysis of the H. amorginum extracts with a focus on specific metabolites. However, as these techniques were used complementary are not intended to fully reveal the phytochemical profile of the extracts of the Greek Helichrysum taxa. Additional research is certainly needed including:
- Chromatographic techniques such as TLC, prep-TLC, prep-HPLC, Low and Medium Pressure Column Chromatography, Countercurrent chromatography (CCC), Solid phase extraction (SPE), Size Exclusion chromatography (SEC), Affinity Chromatography, Ion Exchange Chromatography
- Spectroscopic techniques such as 1D and 2D NMR (e.g., COSY, HSQC, HMBC), 13C NMR, Fluorescence Spectroscopy, Circular Dichroism (CE) and Hyphenated Techniques such as LC-PDA-MS, LC-NMR, LC-NMR-MS, CE-MS
A concluding paragraph about the limitations of the study was added in the subsection 2.3.
- The conclusions must be rewritten to eliminate methodological aspects that were already described in the methods.
Following the reviewer’s advice, the conclusions have been rewritten in the revised version of the manuscript (see track changes).
Reviewer 2 Report
Comments and Suggestions for Authors
This study aimed to investigate phytochemical profile of the Greek Helichrysum taxa and comparatively evaluate their total polyphenol content, which is helpful for utilization of the local endemic Helichrysum taxa. However, major revision is recommended, and the language expression should be some checked thoroughly, questions and suggestions are as following.
1 the abstract is long, but important research results are not displayed, please reorganized.
2 the keywords should be properly chosen, and delete some non-key words.
3 for the introduction section, the first and second paragraphs should be simplified and reorganized for readability.
4 table 1 should be reorganized, because so many compounds are same in different samples, and you are suggested to provide the essential data of the identified compounds, for example, MS data.
5 for section 2.3, the discussion for UV-vis of the extracts and the probable compounds is unconvincing in line 210-229, because the extract is just crude extract without any purification, so many unknown impurities may contribute the λmax, for example, soluble proteins and amino acids.
6 figure 2-4 is meaningless, the Neochlorogenic acid (NCA) and chlorogenic acid (CA) are typical hydroxycinnamic acids with typical UV-vis characteristic at about 320 nm. Similarly, kaempferol-3-O-glucoside could be measured at the maximum absorption wavelength of typical flavonoid (about 350 nm).
7 for section 3. Materials and Methods, the method for identification of phytochemicals with HPLC-MS is missing, which is important for this research, please provide the equipment and MS conditions.
Author Response
This study aimed to investigate phytochemical profile of the Greek Helichrysum taxa and comparatively evaluate their total polyphenol content, which is helpful for utilization of the local endemic Helichrysum taxa. However, major revision is recommended, and the language expression should be some checked thoroughly, questions and suggestions are as following.
1 the abstract is long, but important research results are not displayed, please reorganized.
We would like to thank the reviewer for this comment. The abstract in the revised version of the manuscript has been rewritten as advised (see track changes).
2 the keywords should be properly chosen, and delete some non-key words.
Non-relative keywords were deleted in the revised version of the manuscript as suggested by the reviewer (see track changes).
3 for the introduction section, the first and second paragraphs should be simplified and reorganized for readability.
We would like to thank the reviewer for this comment. The first two paragraphs in the introduction section of the revised manuscript have been rearranged as advised (see track changes).
4 table 1 should be reorganized, because so many compounds are same in different samples, and you are suggested to provide the essential data of the identified compounds, for example, MS data.
We would like to thank the reviewer for this comment. The changes suggested have been made in the revised version of the manuscript as required (see track changes). The content of Table 1 has been reorganized, and the MS data have been added to the Supplementary materials (see revised supplementary materials).
5 for section 2.3, the discussion for UV-vis of the extracts and the probable compounds is unconvincing in line 210-229, because the extract is just crude extract without any purification, so many unknown impurities may contribute the λmax, for example, soluble proteins and amino acids.
The crude extracts used in this study were not purified, and therefore, the presence of unknown impurities may have influenced the UV-vis absorption spectra. These impurities could include soluble proteins, amino acids, and other organic compounds that absorb at similar wavelengths. The presence of impurities can lead to overlapping absorption peaks, making it challenging to accurately attribute the observed λmax values to specific compounds. For example, the absorption peaks at 266 nm, 289 nm, and 321 nm for H2O (STE) and 265 nm, 286 nm, and 322 nm for MeOH:H2O (7:3, STE) could be influenced by both the target compounds and impurities. Techniques such as solid-phase extraction (SPE) or liquid-liquid extraction (LLE) can be employed to remove impurities and isolate the target compounds. This would help in obtaining clearer and more reliable UV-vis spectra. In future studies we will consider using purified extracts to minimize the impact of impurities on the UV-vis absorption spectra. Additionally, complementary analytical techniques such as high-performance liquid chromatography (HPLC) coupled with mass spectrometry (MS) will be used to confirm the identity and purity of the compounds.
6 figure 2-4 is meaningless, the Neochlorogenic acid (NCA) and chlorogenic acid (CA) are typical hydroxycinnamic acids with typical UV-vis characteristic at about 320 nm. Similarly, kaempferol-3-O-glucoside could be measured at the maximum absorption wavelength of typical flavonoid (about 350 nm).
The Figures provide valuable information on the efficiency of different extraction methods, quantitative differences in compound concentrations, the presence of additional compounds, and the optimization of extraction conditions. However, Figures 2-4 have been moved to the supplementary material file.
7 for section 3. Materials and Methods, the method for identification of phytochemicals with HPLC-MS is missing, which is important for this research, please provide the equipment and MS conditions.
The LC-HRMS/MS method, the equipment and MS conditions applied in this investigation have been added to the revised main text of the manuscript as suggested by the reviewer.
Reviewer 3 Report
Comments and Suggestions for Authors
Dear Authors,
I have carefully reviewed and evaluated your manuscript. Below, you will find my observations, comments and recommendations, which you should take into account in order to update your work accordingly.
Title
The title is clear and reflects the main purpose of the manuscript. It complies with the word limit. Recommendation: No changes are required.
Abstract
Provides a good overview of the study, but includes abbreviations that should be avoided in this section. Recommendation: Rewrite the abstract removing abbreviations (e.g., PLE, TPC assay).
Introduction
Provides adequate context and clearly describes the problem. However, the wording could be optimised to focus more on the knowledge gap and better justify the methodology. Recommendation: 1) Revise the section to reduce information that is not directly related to the objective. 2) Include more recent citations where possible.
Materials and Methods
The description of the techniques employed is clear, but some details could be expanded to ensure full replicability (e.g., specific extraction conditions, equipment configuration). Recommendation: 1) Detail the LC-HRMS conditions and PLE parameters. 2) Include information on how the samples were selected.
Results
The results are well structured and supported by tables and figures. However, some figures could be optimised to facilitate visual interpretation. Recommendation: 1) Revise the graphs to ensure that the axes and legends are completely clear. 2) Include p-values in all statistical comparisons mentioned.
Discussion
The discussion synthesises the results and compares them with previous studies. However, specific hypotheses are missing for future research. Recommendation: 1) Add a concluding paragraph highlighting the limitations of the study. 2) Propose specific hypotheses based on the results obtained.
Conclusion
Clearly responds to the stated objectives. It is concise and aligned with the results. Recommendation: No major changes are required.
References
References are relevant, but some are incomplete (e.g., access dates). Recommendation: Review and complete references according to Plants format.
General Recommendations
1) Review the overall format of the manuscript to ensure that it complies with Plants editorial guidelines.
2) Optimise the language to avoid redundancies and improve clarity.
3) Perform a final spelling and grammar check.
I remain at your disposal for any further clarification. I am grateful for the opportunity to review your work and wish you every success in updating the manuscript.
Regards,
Reviewer
Author Response
Dear Authors,
I have carefully reviewed and evaluated your manuscript. Below, you will find my observations, comments and recommendations, which you should take into account in order to update your work accordingly.
Title
The title is clear and reflects the main purpose of the manuscript. It complies with the word limit. Recommendation: No changes are required.
We are thankful for the kind comments.
Abstract
Provides a good overview of the study, but includes abbreviations that should be avoided in this section. Recommendation: Rewrite the abstract removing abbreviations (e.g., PLE, TPC assay).
The abstract has been rewritten in the revised version of the manuscript as advised by the reviewer (see track changes).
Introduction
Provides adequate context and clearly describes the problem. However, the wording could be optimised to focus more on the knowledge gap and better justify the methodology. Recommendation:
1) Revise the section to reduce information that is not directly related to the objective.
2) Include more recent citations where possible.
We would like to thank the reviewer for this comment. The first two paragraphs in the introduction section of the revised manuscript have been rearranged as advised (see track changes).
Materials and Methods
The description of the techniques employed is clear, but some details could be expanded to ensure full replicability (e.g., specific extraction conditions, equipment configuration). Recommendation:
1) Detail the LC-HRMS conditions and PLE parameters.
The LC-HRMS/MS method, the equipment and MS conditions applied in this investigation have been added to the revised main text of the manuscript as suggested by the reviewer.
2) Include information on how the samples were selected.
As suggested by the reviewer, new information has been added in subsection 3.4 and different subsubsections have been added in the revised version of the manuscript to improve understanding regarding the samples’ selection (see track changes).
Results
The results are well structured and supported by tables and figures. However, some figures could be optimised to facilitate visual interpretation. Recommendation:
1) Revise the graphs to ensure that the axes and legends are completely clear.
2) Include p-values in all statistical comparisons mentioned.
In every figure of the revised version of the manuscript, we have made sure the axes and legends were visible. Moreover, p-values have been incorporated into every statistical comparison to enhance comprehension of the importance of our findings as suggested by the reviewer.
Discussion
The discussion synthesises the results and compares them with previous studies. However, specific hypotheses are missing for future research. Recommendation:
1) Add a concluding paragraph highlighting the limitations of the study.
2) Propose specific hypotheses based on the results obtained.
Additional text has been added as advised by the reviewer highlighting the limitations of the study and proposing specific hypotheses based on the results obtained (see track changes).
Conclusion
Clearly responds to the stated objectives. It is concise and aligned with the results. Recommendation: No major changes are required.
Following the advice given by the first reviewer, the conclusions have been improved in the revised version of the manuscript (see track changes).
References
References are relevant, but some are incomplete (e.g., access dates). Recommendation: Review and complete references according to Plants format.
The references have been carefully reviewed again, and some discrepancies have been corrected in the revised manuscript (see track changes); they will be finalized during the copy-edit and proofreading processes upon acceptance of this manuscript.
General Recommendations
1) Review the overall format of the manuscript to ensure that it complies with Plants editorial guidelines.
The formatting has been carefully reviewed again, and some discrepancies have been corrected in the revised manuscript (see track changes); this will be finalized during the copy-edit and proofreading processes upon acceptance of this manuscript.
2) Optimise the language to avoid redundancies and improve clarity.
Language has been carefully reviewed again, and some discrepancies have been corrected in the revised manuscript (see track changes). The English check of the final version will be finalized during the copy-edit and proofreading processes upon acceptance of this manuscript.
3) Perform a final spelling and grammar check.
English spelling has been carefully reviewed again, and some discrepancies have been corrected in the revised manuscript (see track changes). The English check of the final version will be finalized during the copy-edit and proofreading processes upon acceptance of this manuscript.
I remain at your disposal for any further clarification. I am grateful for the opportunity to review your work and wish you every success in updating the manuscript.
Regards,
Reviewer
Round 2
Reviewer 1 Report
Comments and Suggestions for Authors
The authors have adequately addressed the major concerns raised in the initial review.
Reviewer 2 Report
Comments and Suggestions for Authors
The authors have addressed all my comments, and improved the manuscript greatly by detailed revisions. I have no further question.